# EBNA1 IgM-Based Discrimination Between Rheumatoid Arthritis Patients, Systemic Lupus Erythematosus Patients and Healthy Controls

**DOI:** 10.3390/antib8020035

**Published:** 2019-06-01

**Authors:** Nicole Hartwig Trier, Anette Holck Draborg, Louise Sternbæk, Lone Troelsen, Janni Lisander Larsen, Søren Jacobsen, Gunnar Houen

**Affiliations:** 1Department of Autoimmunology, Statens Serum Institut; Oerestads boulevard 5, 2300 Copenhagen S, Denmark; nhp@ssi.dk (N.H.T.); ahdraborg@gmail.com (A.H.D.); louise.sternbeak@phiab.se (L.S.); 2Department of Clinical Immunology, Rigshospitalet, Copenhagen University Hospital, Blegdamsvej 9, 2100 Copenhagen, Denmark; lone.troelsen@gmail.com; 3Copenhagen Lupus and Vasculitis Clinic, Center for Rheumatology and Spine Diseases, Rigshospitalet, Blegdamsvej 9, 2100 Copenhagen, Denmark; jannilisander@icloud.com (J.L.L.); soren.jacobsen.01@regionh.dk (S.J.); 4Department of Biochemistry and Molecular Biology, University of Southern Denmark, Campusvej 55, 5230 Odense M, Denmark

**Keywords:** Epstein–Barr Virus, EBNA1, EAD, IgM, IgA, IgG, sensitivity, specificity, rheumatoid arthritis, systemic lupus erythematosus

## Abstract

Epstein–Barr Virus (EBV) has been associated with development of rheumatic connective tissue diseases like rheumatoid arthritis (RA) and systemic lupus erythematosus (SLE) in genetically susceptible individuals. Diagnosis of RA and SLE relies on clinical criteria in combination with the presence of characteristic autoantibodies. In addition, antibodies to several EBV antigens have been shown to be elevated in patients with these diseases compared to healthy controls (HC). Here, we elaborated improved enzyme-linked immunosorbent assays for antibodies (IgM, IgA, IgG) to the EBV proteins Epstein-Barr Virus nuclear antigen (EBNA)1 and early antigen diffuse (EAD) in order to determine their potential diagnostic role. We showed that especially EBNA1 IgM distinguished RA from SLE and HCs and also distinguished SLE from HCs. EBNA1 IgA was almost as effective in differentiating RA from SLE and HC, while EAD IgG and IgA were able to discern SLE patients from RA patients and HCs. Collectively, these findings illustrate the potential diagnostic use of antibodies to EBV proteins to diagnose RA and to differentiate SLE from RA.

## 1. Introduction

Epstein-Barr virus (EBV), is one of the most common human viruses to encounter. As a consequence, up to 99% of the world’s population is infected at some stage during their lifetime [1,2,3]. The majority of the population is infected during early childhood (up to 95%) [1,2,3]. Primary infection with EBV is usually mild but may give rise to infectious mononucleosis (IM), a condition with fever, excessive production of lymphocytes and swollen nasopharyngeal lymph nodes, in adolescents [4]. Besides IM, EBV has been associated with a broad range of diseases including lymphomas, nasopharyngeal cancer, gastric cancer, multiple sclerosis, rheumatoid arthritis (RA) and systemic lupus erythematosus (SLE). However, the precise role of EBV in the etiology of these diseases remains unknown.

RA is a chronic autoimmune disease with inflammation of joints and multiple systemic symptoms [5,6]. Characteristic antibodies are rheumatoid factors (RFs) and citrulline-dependent antibodies, referred to as anti-citrullinated protein antibodies (ACPAs) [5,7]. RA is most prevalent in women and is caused by a combination of genetic and environmental factors [5,8]. Among multiple genetic factors, especially major histocompability complex (MHC) II alleles with specific amino acid motifs, collectively called the shared epitope (SE), strongly predispose to the disease, and especially in ACPA-positive individuals [8,9,10]. Among the environmental factors, vitamin D, smoking, obesity in adolescence and EBV infection are most influential. 

SLE is another female predominant prototype autoimmune disease characterized by severe fatigue, skin rash, and other symptoms [11,12,13]. The autoantibody spectrum in SLE is extremely diverse, but characteristic autoantibodies are antibodies to double-stranded DNA and ribonucleoprotein (RNP) [11,12,13,14,15]. Multiple genetic factors contribute to the development of SLE, but certain MHC II alleles are strongly predisposing albeit not the same as in RA [16]. Environmental factors overlap with those for RA to a large degree, and vitamin D, smoking and EBV infection have a major impact on disease development [11,17,18,19,20].

The relation between SLE and EBV has been suggested for decades, and it has been shown that SLE patients have defective/lower levels of EBV-specific T cells but increased levels of EBV antibodies, particularly to the lytic early antigen diffuse (EAD) [21,22,23,24,25]. Similarly, the connection between RA and EBV has been suspected for a long time, but the EBV antibody profile appears to be somewhat different in comparison with SLE.

Here, we have investigated several isotypes of EBV antibodies in cohorts of RA and SLE patients using an improved enzyme-linked immunosorbent assay (ELISA) setup and show that Epstein–Barr virus nuclear antigen (EBNA) 1 IgM discriminates efficiently between RA patients, SLE patients and healthy controls (HC)s. These results may have diagnostic value and further testify to the possible etiological role of EBV in these diseases.

## 2. Materials and Methods

### 2.1. Materials

Tris, diethanolamine, alkaline phosphatase (AP)-conjugated goat immunoglobulins to human IgM, IgA and IgG (GaHIgM/A/G^AP^) and para-nitrophenyl phosphate (pNPP) substrate tablets were from Sigma (St. Louis, MO, USA). MgCl_2_, Tween 20, NaHCO_3_, Na_2_CO_3_, were from Merck (Darmstadt, Germany). NUNC MaxiSorp and PolySorp plates were from Thermo Fisher Scientific (Roskilde, Denmark). 

EBV proteins, EBNA1 (recombinant, mosaic) (Escherichia coli-derived, EBV271, purity >95%) and EAD (recombinant) (Escherichia coli-derived, EBV-272, purity >95%), were purchased from Prospec Protein Specialist (Ness-Ziona, Israel).

### 2.2. Patient Sera

RA patient sera were collected at the Department of Rheumatology, Frederiksberg Hospital (Copenhagen, Denmark). SLE patient sera were collected at the Center for Rheumatology, Rigshospitalet, Copenhagen, Denmark. HC sera were obtained from volunteers at Rigshospitalet and at Statens Serum Institut (Copenhagen, Denmark). Twenty-nine SLE sera were enrolled of which 28 were from females. Average age for SLE patients was 36 years. HCs were gender and age matched according to the SLE patients. Twenty-nine HCs were enrolled of which 27 were female. The average age for the HC individuals was 41 years. Seventy-seven RA sera were enrolled in the study, 76% of these were from females. The average age was 56 years for the RA individuals. 

### 2.3. Ethics

All samples were obtained with written consent. The study was conducted in accordance with the relevant ethical guidelines and was approved by the Scientific Ethical Committee of the Capital Region, Denmark H-A-2007-0114), project ID:19980024 PMC and H-15009640).

### 2.4. Enzyme-Linked Immunosorbent Assay

Polysorp plates were coated overnight at 5 °C with EBNA1 and EAD antigens (1 µg/mL) in 50 mM sodium carbonate buffer, pH 9.6 (100 µL/well). Then, the wells were washed and blocked with Tris-Tween-NaCl (TTN) buffer (50 mM Tris, pH 7.5, 1 % Tween 20, 0.3 M NaCl,) by 3 × 5 min incubation on a shaking table and using 200 µL/well. Sera were diluted 1:10 (IgM, IgA) or 1:100 (IgG) in TTN buffer, added to the microtitre plates, in both coated and non-coated wells in duplicates, and incubated for 1 h followed by 3 washes as described above. Secondary antibodies GaHIgM/A/G^AP^ (one for each plate) were diluted 1:2000 in TTN and added to wells (100 µL/well) and incubated 1 h followed by another 3 washes with TTN. Finally, 100 µL AP substrate solution (10 mg pNPP substrate tablet per 10 mL AP buffer (1 M diethanolamine, 0.5 mM MgCl_2_, pH 9.8)) was added to all wells. Absorbances were recorded on a Versamax microplate reader (Molecular Devices, Sunnyvale, CA, USA) using a wavelength of 405 nm with background subtraction at 650 nm. 

The sample absorbance values of non-coated wells were subtracted from coated wells after averaging the duplicates. A standard curve (1:10, 1:20, 1:40, 1:80, 1:160, 1:320) was included for each assay using a pool of high-titre sera samples and all absorbance values were normalized relative to this standard (EBNA1 or EAD).

### 2.5. Statistics

Each sample was analysed for antibody reactivity in duplicates. Statistical analyses, generation of receiver operating curves (ROCs) and calculation of area under the curves (AUCs) were performed using GraphPad Prism software (GraphPad, San Diego, CA, USA). For determination of significance, Dunnett’s test was performed, which compares all columns to control columns. 

## 3. Results

### Determination of Antibody Titers to Early Antigen Diffuse and Epstein–Barr Virus Nuclear Antigen 1 by Enzyme-Linked Immunosorbent Assay 

Sera from SLE, RA and HC individuals were tested for antibody reactivity (IgM, IgA, IgG) to EAD and EBNA1 by ELISA. Analysis of antibodies to EBNA1 revealed high levels of IgM and IgA in RA patient sera followed by SLE patient sera and HC sera (Figure 1).

For EBNA1 IgM and IgA, the differences were highly statistically significant between RA and SLE and RA and HCs (*p* < 0.0001) (Figure 1a,b). Moreover, ROC curves (Figure 2) revealed high sensitivities and specificities with AUC values > 0.9 (Table 1). Best discrimination between RA and HC was achieved by EBNA1 IgM (AUC = 0.995), which also discriminated RA from SLE (AUC = 0.953) and SLE from HC (AUC = 0.916). The corresponding values for EBNA1 IgA were slightly lower but still very high (Table 1). In contrast EBNA1 IgG had much lower discriminating power (Figure 1, Appendix A) with AUC values ranging from 0.5198–0.8356, which also is reflected in the antibody reactivity pattern (Figure 1c). 

Analysis of antibodies to EAD revealed low levels of IgA, whereas antibody titers for EAD IgM and IgG were increased (Figure 1d–f). No evident reactivity pattern was observed compared to EBNA1. For EAD IgA and IgG, the difference between RA and SLE and SLE and HC was statistically significant, whereas for EAD IgM, a statistically significant difference was observed between RA and SLE and RA and HC. EAD IgM had lower discriminating power compared to EBNA1 IgM, but EAD IgA and IgG had fairly good ability to discriminate SLE from HC with AUCs of 0.827 and 0.835, respectively (Table 1). Combinations of EBNA1 and EAD antibodies were also evaluated (e.g., EAD/EBNA1, Appendix A), but none had better discriminating power than EBNA1 IgM.

## 4. Discussion

ELISA is one of the most important techniques for antibody-based diagnostics. Polystyrene plates are commonly used for ELISAs relying on immobilization of antigens by physical adsorption, and plates with high binding capacity are often used (e.g., Maxisorp plates), however, such plates often yield excessive non-specific binding, which may overshadow specific signals [26,27]. Less hydrophobic plates (e.g., Polysorp) have lower binding capacity but in general give fewer problems with non-specific binding. In previous studies, Maxisorp plates were primarily used without standards, yielding only absorbances as a measure of antibody reactivity [28,29]. Here, we used Polysorp plates in combination with a standard to determine relative concentrations of IgM, IgA and IgG to EBNA1 and EAD. The use of Polysorp plates, however, requires higher amounts of sera and longer development times, which, nevertheless, is compensated by better results. 

IgM and IgA to EBNA1 determined by the ELISA elaborated here showed a very good ability to discriminate RA from SLE samples and both from HCs. For EAD, IgG and IgA could discriminate fairly well between SLE samples and RA patients or HCs. This is in good agreement with many other reports in the literature [23,25,30,31,32,33,34].

The serological diagnostics of RA and SLE rely to a large part on determination of RFs and ACPAs for RA and dsDNA antibodies and anti-nuclear antibodies (ANA) for SLE [5,7,11,12,13,14,15,35]. ELISA is a main technique for determination of these autoantibodies, but ANAs are often determined by indirect immunofluorescence with HEp-2 cells as substrate [36,37]. Based on the results obtained here, it is tempting to speculate that most of these ELISAs could be improved by using microtiter plates with low non-specific binding. 

Clinical utility depends on assay sensitivity and specificity [38]. For the RF and ACPA analyses used in serological RA diagnostics these are typically in the range 70–95%, and the same applies for ANA and dsDNA antibodies in SLE diagnostics [35,39]. Since EBV infection has been shown to be an important environmental factor in triggering RA and SLE, antibodies to EBV antigens would be expected to have diagnostic value, although this would also be expected to be of limited value due to the infection of a major part of the human population by this virus.

Previous studies have, for a major part, shown elevated levels of EBV antibodies in RA and SLE patients, however, with different antibody profiles. In RA patients, VCA IgG, EAD IgG, EBNA1 IgG have been found to be elevated in a few but not all studies, and with low specificities [31,33,38,40,41,42,43].

In SLE patients, the following EBV antibodies have generally been found to be elevated: VCA IgM, IgA, IgG, EAD IgM, IgA, IgG, EBNA1 IgA [17,18,34,42,43,44,45,46,47,48,49]. A basic feature of EBV lifecycle is the ability of the virus to shift between (memory) B cell and epithelial cell infection. Thus, the findings described above are in accordance with that in SLE, Igs to EAD, and in particular IgA, primarily are related to epithelial tissue infection, where disease activity protrudes, whereas in RA, Igs to EBNA-1, and in particular IgM, primarily are associated with infected B-cells in the inflamed joints.

A major problem in the interpretation and comparison of the results mentioned above is a lack of correction for non-specific binding, which is a major problem in ELISA with sera from patients with autoimmune connective tissue disease [39,40]. In this study, an improved ELISA system for detection and quantification of EBV antibodies was used with systematic correction for non-specific binding, even if this was very low with the Polysorp plates. In order to do so, Polysorp plates were applied with a systematic use of an internal standard curve, followed by the subtraction of background levels determined by serum reactivity in uncoated wells. This is in contrast to many previous studies, where antibody levels were determined as relative absorbances using the more non-specific binding Maxisorp microtitre plates [28,29]. The application of Polysorp microtiter plates yielded highly sensitive and specific assays. Using this ELISA system, it was found that EBNA1 IgM had high sensitivity and specificity for RA and SLE. This would seem to point to an important etiological role of EBV in these diseases, a theory which is reinforced by the recent finding that antibodies to a single strain-specific citrullinated EBNA2 peptide has a diagnostic sensitivity and specificity for RA comparable to or even better than the currently used commercial assays using pools of citrullinated peptides [29].

As presented, EBNA1 IgM may constitute a supplementary assay to determination of RA. Nevertheless, it remains to be determined how powerful this setup is to differentiate the remaining connective tissues diseases from each other, i.e., is this image specific for RA? And what about Sjogren’s syndrome and systemic sclerosis? This remains to be determined in future studies. Detection of EBV antibodies is most naturally restricted to diseases, where EBV somehow is involved in the disease course.

Collectively, these findings describe the power of EBNA1 IgM to discriminate RA-positive individuals from individuals with SLE and HCs, which ultimately may contribute to the generation of new and simplified diagnostic assays. 

## Figures and Tables

**Figure 1 antibodies-08-00035-f001:**
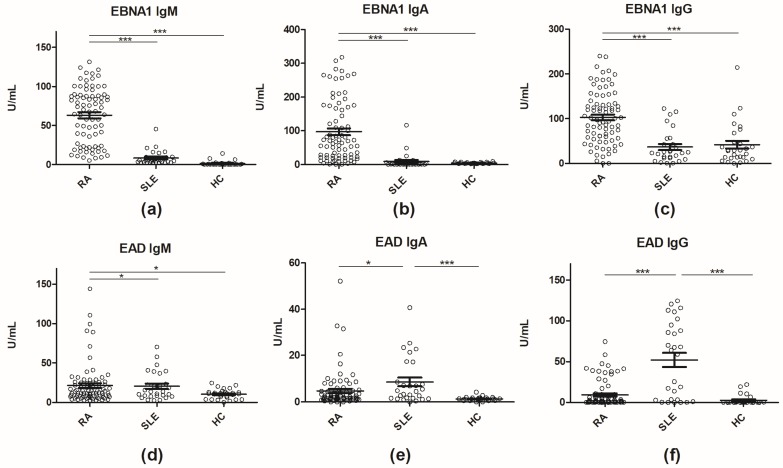
Reactivity of rheumatoid arthritis (RA) (*n* = 77), systemic lupus erythematosus (SLE) (*n* = 27 and healthy control (HC) sera (*n* = 29) to early antigen diffuse (EAD) and Epstein-Barr Virus nuclear antigen (EBNA) 1 (IgM, IgA and IgG reactivity) analyzed by enzyme-linked immunosorbent assay. (**a**) IgM antibody reactivity to EBNA 1; (**b**) IgA antibody reactivity to EBNA1; (**c**) IgG antibody reactivity to EBNA1; (**d**) IgM antibody reactivity to EAD; (**e**) IgA antibody reactivity to EAD; (**f**) IgG antibody reactivity to EAD. * = *p* < 0.05, *** = *p* < 0.0001. EBNA1, Epstein-Barr virus nuclear antigen 1; EAD, early antigen diffuse; HC, healthy control; RA, rheumatoid arthritis; SLE, systemic lupus erythematosus.

**Figure 2 antibodies-08-00035-f002:**
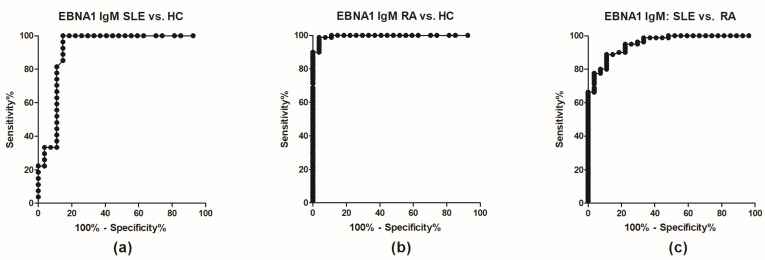
ROC curves for Epstein-Barr nuclear antigen 1 IgM in systemic lupus erythematosus (SLE), rheumatoid arthritis (RA) and healthy control (HC) sera. (**a**) SLE vs HC; (**b**) RA vs HC; (**c**) SLE vs RA. Epstein-Barr virus nuclear antigen 1; HC, healthy control; RA, rheumatoid arthritis; SLE, systemic lupus erythematosus.

**Table 1 antibodies-08-00035-t001:** IgG, IgA, IgM antibodies to EAD and EBNA1 in SLE (*n* = 27), RA (*n* = 77) and HC (*n* = 29). Overview of the best discrimination test with AUC > 0.8.

	AUC	Std. Error	95% Confidence Interval	*p* Value
*EBNA1 IgG*				
SLE vs HC	0.5198	0.7885	0.3653–0.6744	0.8006
RA vs HC	0.8125	0.04936	0.7157–0.9093	<0.0001
SLE vs RA	0.8356	0.04357	0.7503–0.921	<0.0001
*EBNA1 IgA*				
SLE vs HC	0.7037	0.08256	0.5419–0.8655	0.0102
RA vs HC	0.9456	0.0237	0.8992–0.992	<0.0001
SLE vs RA	0.9183	0.03222	0.8551–0.9814	<0.0001
*EBNA1 IgM*				
SLE vs HC	0.9156	0.04432	0.8288–1.003	<0.0001
RA vs HC	0.9954	0.04285	0.987–1.004	<0.0001
SLE vs RA	0.953	0.01944	0.9149–0.9911	<0.0001
*EAD IgG*				
SLE vs HC	0.8347	0.05862	0.7298–0.9496	<0.0001
RA vs HC	0.7377	0.06006	0.62–0.8554	0.0002
SLE vs RA	0.7477	0.0694	0.6117–0.8837	<0.0001
*EAD IgA*				
SLE vs HC	0.8272	0.06039	0.7088–0.9455	<0.0001
RA vs HC	0.7169	0.04997	0.619–0.8248	0.0008
SLE vs RA	0.6544	0.06353	0.5299–0.7789	0.0410
*EAD IgM*				
SLE vs HC	0.6612	0.07527	0.5137–0.8087	0.0421
RA vs HC	0.6306	0.05724	0.5184–0.7427	0.0431
SLE vs RA	0.5417	0.06394	0.4163–0.667	0.8694

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
