# Peer review of "EBNA1 IgM-Based Discrimination Between Rheumatoid Arthritis Patients, Systemic Lupus Erythematosus Patients and Healthy Controls"

_2073-4468, 2019, doi:10.3390/antib8020035_

Reviewer 1 Report

The visibility of error bars in Fig. 1 can be imroved. An option may be the use of partially unfilled circles, squares and triangles.

The legend of Fig. 1 features HD (healthy donor) while otherwise exclusively HC is used. This should be adjusted or explained.

Author Response

Amended as requested. The visibility of the error bars in Fig. 1 has been improved. HD has been replaced with HC in Fig. 1.  

Reviewer 2 Report

The paper “ EBNA1 IgM-based discrimination between rheumatoid arthritis patients, systemic 2 lupus erythematosus patients and healthy controls “ reports on an improved ELISA test to discriminate patients affected by rheumatoid arthritis and systemic lupus erythematosus from healthy controls by the use of Epstein Barr Virus proteins EBNA1 and EAD as immunoreactive antigens. The work is very interesting, scientifically sound and data are well presented, however, the manuscript can be improved by the following revisions:

1)    Line 39-40: please rephrase the sentence, it is too colloquial

2)    Please provide information on EBNA1 and EAD antigens: recombinant? Purity? PDBB access number

3)    Discussion session line 162-164: please elaborate. What are the specific protocol changes that  provided better results? Please specify and compare protocols used with the two different plates

4)    Line 168: add references

5)    Line 171: define IFF

6)    Line 189-190: please elaborate and specify the corrections used to improve results

7)    Can you speculate a reason for slightly different diagnostic performance of the different antibody isotypes in SLE and RA? Please add to the discussion section

Author Response

1)      Line 39-40: please rephrase the sentence, it is too colloquial

Amended as requested. The sentence has been rephrased.

2)      Please provide information on EBNA1 and EAD antigens: recombinant? Purity? PDBB access number

Amended as requested. The origin of the coated antigens has been elaborated in the methods section.

3)    Discussion session line 162-164: please elaborate. What are the specific protocol changes that provided better results? Please specify and compare protocols used with the two different plates

This has been elaborated in the discussion. Previous studies using Maxisorp plates were in many cases conducted without the use of a standard, yielding absorbances as a measure of antibody reactivity. The current studies were conducted using Polysorp plates, which have much lower non-specific binding and therefore are more sensitive. Moreover, an internal standard was applied, which allowed quantification of relative antibody concentrations in U/mL. 

3)      Line 168: add references

Amended as requested. A basic feature of EBV lifecycle is the ability of the virus to shift between (memory) B cell and epithelial cell infection. We suggest that in RA, EBNA1 IgM is primarily related to (memory) B cell infection in joints, whereas in SLE, EAD IgA is related to epithelial infection in affected tissues.

5)    Line 171: define IFF

Amended as requested.

6)    Line 189-190: please elaborate and specify the corrections used to improve results.

Amended as requested.

7)    Can you speculate a reason for slightly different diagnostic performance of the different antibody isotypes in SLE and RA? Please add to the discussion section.

Amended as requested.

Reviewer 3 Report

The authors have developed an improved ELISA for the detection of EBV proteins EBNA1 and EAD. Their finding could potentially lead to new approach needed for rheumatoid arthritis (RA) and systemic lupus erythematosus (SLE) diagnosis. However, the data is still preliminary and therefore warrants addition studies in the future. Furthermore, the manuscript may still benefit from minor language proofreading.

 Below are additional comments that the authors should address.

 Line 41: include appropriate reference

Line 97-108: This method section needs additional information in order to improve clarity. E.g what were the standard used? How did you distinguish btw IgM, IgA and IgG.

Line 101-102: According the sentence, it is not clear if sera were actually added to the wells?

Line 120: It is not clear how the IgM, IgA, IgG to EAD and EBNA1 were generated and determined?

Discussion

The potential limitations of using these antibodies as a diagnostic tool should be further discussed.

Author Response

Line 41: include appropriate reference

Amended as requested.

Line 97-108: This method section needs additional information in order to improve clarity. E.g what were the standard used? How did you distinguish btw IgM, IgA and IgG.

The ELISA section has been elaborated. For generation of a standard curve from EBNA1 and EAD was used in combination with a positive high titre serum pool. Only one secondary antibody was added to each microtitre plate, hence each plate yielded specific results for IgA, IgM and IgG. This has been added to the section. 

Line 101-102: According the sentence, it is not clear if sera were actually added to the wells?

The sentence has been rephrased.

Line 120: It is not clear how the IgM, IgA, IgG to EAD and EBNA1 were generated and determined?

The sentence has been rephrased.

Discussion

The potential limitations of using these antibodies as a diagnostic tool should be further discussed.

Amended as requested.

Round  2

Reviewer 3 Report

no further comments